# Immunolocalization of Enzymes/Membrane Transporters Related to Bone Mineralization in the Metaphyses of the Long Bones of Parathyroid-Hormone-Administered Mice

**DOI:** 10.3390/medicina59061179

**Published:** 2023-06-20

**Authors:** Takahito Mae, Tomoka Hasegawa, Hiromi Hongo, Tomomaya Yamamoto, Shen Zhao, Minqi Li, Yutaka Yamazaki, Norio Amizuka

**Affiliations:** 1Department of Developmental Biology of Hard Tissue, Graduate School of Dental Medicine, Faculty of Dental Medicine, Hokkaido University, Sapporo 060-8586, Japanhiromi@den.hokudai.ac.jp (H.H.); tomomaya@den.hokudai.ac.jp (T.Y.); amizuka@den.hokudai.ac.jp (N.A.); 2Department of Gerontology, Graduate School of Dental Medicine, Faculty of Dental Medicine, Hokkaido University, Sapporo 060-8586, Japan; 3Northern Army Medical Unit, Camp Makomanai, Japan Ground Self-Defense Forces, Sapporo 005-8543, Japan; 4Department of Endodontics and Operative Dentistry, Shanghai Ninth People’s Hospital, College of Stomatology, Shanghai Jiao Tong University School of Medicine, Shanghai 200025, China; 5Center of Osteoporosis and Bone Mineral Research, Department of Bone Metabolism, School of Stomatology, Shandong University, Jinan 250012, China; liminqi@sdu.edu.cn

**Keywords:** mineralization, parathyroid hormone, ENPP1, TNALP, PHEX, MEPE

## Abstract

The present study aimed to demonstrate the immunolocalization and/or gene expressions of the enzymes and membrane transporters involved in bone mineralization after the intermittent administration of parathyroid hormone (PTH). The study especially focused on TNALP, ENPP1, and PHOSPHO1, which are involved in matrix vesicle-mediated mineralization, as well as PHEX and the SIBLING family, which regulate mineralization deep inside bone. Six-week-old male mice were subcutaneously injected with 20 μg/kg/day of human PTH (1–34) two times per day (*n* = 6) or four times per day (*n* = 6) for two weeks. Additionally, control mice (*n* = 6) received a vehicle. Consistently with an increase in the volume of the femoral trabeculae, the mineral appositional rate increased after PTH administration. The areas positive for PHOSPHO1, TNALP, and ENPP1 in the femoral metaphyses expanded, and the gene expressions assessed by real-time PCR were elevated in PTH-administered specimens when compared with the findings in control specimens. The immunoreactivity and/or gene expressions of PHEX and the SIBLING family (MEPE, osteopontin, and DMP1) significantly increased after PTH administration. For example, MEPE immunoreactivity was evident in some osteocytes in PTH-administered specimens but was hardly observed in control specimens. In contrast, mRNA encoding cathepsin B was significantly reduced. Therefore, the bone matrix deep inside might be further mineralized by PHEX/SIBLING family after PTH administration. In summary, it is likely that PTH accelerates mineralization to maintain a balance with elevated matrix synthesis, presumably by mediating TNALP/ENPP1 cooperation and stimulating PHEX/SIBLING family expression.

## 1. Introduction

Bone matrix primarily comprises calcium phosphate crystals (hydroxyapatite) and type I collagen; therefore, bone formation can be divided into two aspects: the synthesis of organic bone matrices, including collagens and non-collagenous proteins, and the mineralization of the bone matrix. Bone mineralization, i.e., calcium phosphate crystals deposited onto the collagen fibrils, provides hardness, while type I collagen gives flexibility to the bone. Bone mineralization is biologically regulated by osteoblasts that secrete matrix vesicles, small extracellular vesicles that initiate the mineralization process in bones, i.e., matrix vesicle-mediated mineralization [1,2,3,4,5]. To allow matrix vesicle-mediated mineralization, the inflow of abundant calcium ions (Ca^2+^) and inorganic monophosphate ions or Pi ions (PO_4_^3−^) into the matrix vesicles is imperative to nucleate calcium phosphate crystals. The crystalline calcium phosphates grow inside the matrix vesicles and penetrate the plasma membranes of the vesicles to form mineralized nodules, which are globular assemblies of fine needle-like calcium phosphate crystals [1,2,5].

Important aspects of bone mineralization include biological action on the nucleation and growth of crystalline calcium phosphates and Ca^2+^ and PO_4_^3−^ supplementation. There are numerous free Ca^2+^ ions in tissue fluid surrounding the matrix vesicles to support the direct inflow of Ca^2+^ into the vesicles by Ca^2+^-ATPase [2]. In addition, mature osteoblasts and matrix vesicles have several membrane transporters and enzymes involved in local Pi supplementation. One of the most important enzymes that can initiate mineralization in bone is tissue-nonspecific alkaline phosphatase (TNALP), which can hydrolyze various phosphate esters, especially pyrophosphate (PPi), and therefore, it is responsible for the local production of Pi in bone [6,7,8,9,10]. Another putative inducer of mineralization is phosphoethanolamine/phosphocholine phosphatase (PHOSPHO1) [11,12,13,14], which is thought to function inside cells and matrix vesicles, catalyzing the division of phosphocholine, a constituent of the plasma membrane, into choline and Pi. In contrast to these mineralization-inducing enzymes, ectonucleotide pyrophosphatase/phosphodiesterase 1 (ENPP1) is thought to inhibit mineralization [15,16,17,18]. Crystalline structure analysis has demonstrated that extracellular adenosine triphosphate (ATP) is accommodated in a pocket formed by an insertion loop of ENPP1, explaining the preference of ENPP1 for an ATP substrate [19]. Additionally, ankylosis (ANK), a non-enzymatic plasma membrane PPi channel, might allow intracellular PPi to pass through the cell membrane from the cytoplasm to the outside of the cell [20,21]. The resultant extracellular PPi can serve as an inhibitor of mineralization to avoid pathological mineralization [21]. However, in bone, extracellular PPi is hydrolyzed by TNALP to Pi, which is used for the constituents of mineral crystals. Therefore, the finely tuned cooperation/activities of these enzymes and membrane transporters in the osteoid, which is the site of primary mineralization in bone, appears to be necessary for normal mineralization in bone.

Bone mineralization is not only initiated by matrix vesicles-mediated mineralization but also regulated by bone matrix proteins secreted by osteoblasts and osteocytes. Fibroblast growth factor 23 (FGF23) is expressed in bone, especially in osteocytes, and it circulates to the renal proximal tubules to bind the FGFR1c/αklotho receptor complex in order to reduce the reabsorption of Pi in the kidneys [22,23,24,25,26]. However, recent studies have proposed an autocrine function of FGF23, which, at least in part, is related to bone mineralization. Osteoblasts/osteocytes express not only FGF23 but also the FGFR1c/αklotho receptor complex [27,28], consequently regulating bone mineralization in an autocrine manner [29]. In addition, PHEX has been postulated to bind the small integrin-binding ligand N-linked glycoprotein (SIBLING) family including osteopontin, matrix extracellular phosphoglycoprotein (MEPE), dentin matrix protein 1 (DMP1), dentin sialoprotein, and bone sialoprotein for the local regulation of mineralization in bone [30,31,32,33,34,35].

Parathyroid hormone (PTH) (1–34) is a calcitropic hormone that stimulates osteoclastic bone resorption and osteoblastic bone formation. Although the intermittent administration of hPTH (1–34) increases bone volume [36,37], whether it elevates the expression of genes encoding enzymes and membrane transporters associated with bone mineralization is still veiled. Therefore, studying the influence of PTH in vivo on the expression of these genes would provide novel insight into bone mineralization mechanisms.

To estimate bone mineralization, microCT analyses were performed on mice and rats, which provided information regarding mineral contents and bone mineral density (BMD), as well as bone volume and geometrical structure, not only at a macroscale of the whole bone but also at each part of the bone. A long bone is roughly composed of an outer cortical bone and a spongy inner bone. The extremity of an adult long bone is the epiphyses, while young adult mice/rats contain growth plate cartilage between the proximal epiphyses and distal metaphyses. The region of the midshaft is the diaphysis, which features the cortical bone. MicroCT analysis estimates mineral contents and BMD in these parts of long bones at a macroscale. However, to elucidate the PTH effects on the mineralization process and the enzymes/membrane transporters related to bone mineralization, it is necessary to employ microscale observation, localizing these enzymes/transporters on osteoblasts and osteocytes. As shown by Zimmermann et al. [38] and Buccino et al. [39,40], it is reasonable to explore the complexity of multiscale damage processes that make fracture prediction, and it is also necessary to seek an understanding of the origins of disease-related deterioration in a bone’s mechanical properties. Consistently with these reports, bone mineralization appears to be involved in a bone’s mechanical properties.

Therefore, this study aimed to clarify whether PTH may stimulate TNALP/ENPP1 function and the PHEX/SIBLING family to facilitate bone mineralization, independently of the regulation of the serum concentrations of Ca and Pi in the metaphyses of PTH-administered mice.

## 2. Materials and Methods

### 2.1. Animals

Six-week-old male C57BL/6J mice (*n* = 18, Japan CLEA, Tokyo, Japan) were used in this study. The study followed the principles for animal care and research use set by Hokkaido University (approval no.: 15-0032, 20-0019). According to our previous study [37], 20 μg/kg of human PTH (1-34) (Sigma-Aldrich Co., LLC., St. Louis, MO, USA) was subcutaneously injected two times per day (*n* = 6) or four times per day (*n* = 6) into the study mice for 2 weeks. Age-matched mice that underwent vehicle administration were used as controls (*n* = 6).

### 2.2. Preparation of Histochemical Specimens

Before fixation, mice were anesthetized with an intraperitoneal injection of chloral hydrate for body weight determination and blood collection to estimate the concentrations of serum Ca and Pi. The left hind legs of the mice were ligated at the femoral region, and the left tibiae were extracted for analyses involving reverse transcription polymerase chain reaction (RT-PCR) and real-time PCR, as described later. All mice were then perfused with 4% paraformaldehyde diluted in 0.1 M cacodylate buffer (pH 7.4) through the cardiac left ventricle. The right tibiae and femora were immediately removed and immersed in the same fixative for 18 h at 4 °C. The samples were decalcified with a solution of 10% ethylene diamine tetraacetic disodium salt and were dehydrated in ascending ethanol solutions prior to paraffin embedding for immunohistochemical examination. For von Kossa staining, undecalcified tibiae were post-fixed with 1% osmium tetraoxide in a 0.1 M cacodylate buffer for 4 h at 4 °C, dehydrated in ascending acetone solutions, and embedded in epoxy resin (Epon 812, Taab, Berkshire, UK) [41].

### 2.3. Immunostaining of TNALP, ENPP1, PHOSPHO1, FGF23, Osteopontin, and MEPE

Dewaxed paraffin sections were examined for TNALP as previously reported [42,43]. In brief, after inhibition of endogenous peroxidases with methanol containing 0.3% hydrogen peroxidase for 30 min, dewaxed paraffin sections were pretreated with 1% bovine serum albumin (BSA; Serologicals Proteins Inc. Kankakee, IL, USA) in PBS (1% BSA-PBS) for 30 min. Sections were then incubated for 2–3 h at room temperature (RT) with rabbit polyclonal antisera against TNALP [9] at a dilution of 1:300 (1% BSA-PBS). This was followed by incubation with horseradish (HRP)-conjugated anti-rabbit IgG (Jackson Immunoresearch Laboratories Inc., West Grove, PA, USA). For FGF23, after pre-incubation with 1% BSA-PBS for 30 min at RT, sections were incubated with rat anti-FGF23 antibodies (R&D Systems Inc., Minneapolis, MN, USA) at a dilution of 1:100 for 2 h [37] and then incubated with HRP-conjugated anti-rat IgG (Chemicon International Inc., Temecula, CA, USA) at a dilution of 1:100 for 1 h. For ENPP1, sections were incubated with goat polyclonal anti-ENPP1 (Everest Biotech Ltd., Oxfordshire, UK) at a dilution of 1:200 for 1 h and then incubated with HRP-conjugated rabbit anti-goat IgG (American Qualex Scientific Products, Inc., San Clemente, CA, USA). For PHOSPHO1, sections were initially incubated with human monoclonal anti-PHOSPHO1 (Bio-Rad Laboratories Inc., Hercules, CA, USA) at a dilution of 1:100. Subsequently, the sections were incubated with rabbit anti-myc Tag (Medical & Biological Laboratories Co., Ltd., Nagoya, Japan) and then incubated with HRP-conjugated swine anti-rabbit IgG (Agilent Technologies, Inc., Santa Clara, CA, USA) at a dilution of 1:100. For osteopontin and MEPE, dewaxed sections were reacted with rabbit antibodies against mouse osteopontin (LSL Co., Ltd., Tokyo, Japan) at a dilution of 1:2000 or rabbit polyclonal antibodies against MEPE (Cloud-Clone Corp., Katy, TX, USA) at a dilution of 1:100 (1% BSA-PBS) for 2 h. Following several washings in PBS, they were then incubated with HRP-conjugated anti-rabbit IgG (Jackson Immunoresearch Laboratories Inc. West Grove, PA, USA) for 1 h. To visualize all HRP-conjugated immunoreactions, diaminobenzidine tetrahydrochloride was used as a substrate. All sections were counterstained with methyl green and observed under a light microscope (Eclipse Ni, Nikon Instruments Inc., Tokyo, Japan).

### 2.4. Immunofluorescent Staining for the Detection of TNALP and ENPP1

For the detection of TNALP and ENPP1, dewaxed paraffin sections were incubated with 1% BSA-PBS and then with rabbit polyclonal antibodies against TNALP at a dilution of 1:300 (1% BSA-PBS). Subsequently, they were incubated with Alexa Fluor 594-conjugated goat anti-rabbit IgG (Thermo Fisher Scientific, Inc., Waltham, MA, USA) at a dilution of 1:100 (1% BSA-PBS) for 1 h and then with goat polyclonal anti-ENPP1 at a dilution of 1:200 (1% BSA-PBS) after washing with PBS. Additionally, the sections were incubated with Alexa Fluor 488-conjugated donkey anti-goat IgG (Thermo Fisher Scientific, Inc.) at a dilution of 1:100 (1% BSA-PBS) for 1 h. After embedding the sections using VECTASHIELD hard-set mounting medium with DAPI (Vector Laboratories, Inc. Burlingame, CA, USA), the sections were observed under a light microscope.

### 2.5. Von Kossa Staining

Epoxy resin sections from undecalcified specimens were incubated with an aqueous solution of silver nitrate until dark brown/black staining of bone tissue was discernible under a light microscope [44].

### 2.6. Measurement of Serum Ca and Pi

Serum samples were stored at −30°C before the assays. The serum concentrations of Ca and Pi were quantified by Oriental Yeast Co., Ltd. (Tokyo, Japan), using enzymatic methods.

### 2.7. Mineral Appositional Rate after PTH Administration

A 600 × 600 μm region of interest located 150 μm below the growth plate of the tibial metaphysis was used for the assessment of the static parameters of inter-label width (It.L.Th) and inter-label time (It.L.t., 4 days). Then, the mineral appositional rate (MAR) was calculated. Abbreviations and calculations were used according to the recommendations of the ASBMR Histomorphometry Nomenclature Committee, whenever possible [45].

### 2.8. PCR and Real-Time PCR

Left tibiae harvested from control mice and PTH-administered mice, as described above, were immediately frozen in liquid nitrogen and crushed into small pieces. The crushed specimens were homogenized in 10 mL of TRIzol reagent (Life Technologies Co., Carlsbad, CA, USA) per 1 g tissue to extract total RNA. The mixture was centrifuged at 15,000× *g* rpm for 5 min at 4 °C, allowing for the removal of small debris. The supernatant was transferred to a new tube, which was vortexed for 15 s after adding 2 mL of chloroform. The lysate was then transferred to a new tube and was incubated for 5 min at RT. After phase separation, the aqueous phase containing RNA was transferred to a new tube, and RNA was precipitated by adding 5 mL of isopropyl alcohol per 10 mL of TRIzol reagent. After incubation for 10 min at RT, the mixture was centrifuged at 15,000× *g* rpm for 60 min at 4 °C. The resulting RNA pellet was washed with 1 mL of 75% ethanol and was briefly air-dried. The RNA pellet was dissolved in 30 µL of diethyl-pyrocarbonate-treated water. First-strand cDNA was synthesized from 2 µg of total RNA by using the SuperScript VILO cDNA Synthesis Kit (Life Technologies).

The primer sequences used for PCR were as follows: mouse *Gapdh* forward: TGTCTTCACCACCATG GAGAAGG, reverse: GTGGATGCAGGGATGATGTT CTG; mouse *Tnalp* forward: GCCCTCTCCAAGACATATA, reverse: CCATGATCACGTCGATATCC; mouse Enpp1 forward: TATTGGCTATGGACCTGCCTTCAAGC, reverse: GTAGAATCCGGGGCCTCCCGTAG; mouse *Phospho1* forward: GACAATGAGCGGGTGTTTTC, reverse: GGGGATGGTCTCGTAGACAG; mouse *Ank* forward: GAAATCCGGGCTGTCTACCC, reverse: TCAGTGTCATCAGCCATCCAG; mouse *FgfF23* forward: TGTCAGATTTCAAACTCAG, reverse: GGATAGGCTCTAGCAGTG; mouse *Fgfr1c* forward: CTTGACGTCGTGGAAC GATCT, reverse: AGAACGGTCAACCATGCAGAG; mouse *αKlotho* forward: GGGTCACTGGGTCA ATCT, reverse: GCAAAGTAGCCACAAAGG; mouse *Mepe* forward: CCCCAAGAGCAGCAAAGGTA, reverse: CTCCGCTGTGACATCCCTTTA; and mouse *Phex* forward: TTCTGATGGAAGCAGAAACAGGGA, reverse: GGGAATCATAGCGCTGAGTTCTGA. PCR was performed using a thermal cycler (GeneAmp PCR System 2700; Applied Biosystems, Foster City, CA, USA) as follows: denaturation at 94 °C for 30 s; annealing at 60 °C (for *Gapdh, Ank, Phospho1, Phex,* and *Mepe*), 53 °C (for *Fgf23*), 55 °C (for *Fgfr1c* and *αKlotho*), 56°C (for *Tnalp*), or 63 °C (for *Enpp1*) for 30 s; extension at 72 °C for 30 s; and final incubation at 72 °C for 10 min. RT-PCR products were subjected to 2% agarose gel electrophoresis, stained with ethidium bromide, and detected using E-Gel Imager (Life Technologies).

Real-time PCR assays were performed using Taqman probes (Applied Biosystems) for the gene expressions of *Tnalp (Alpl)* (Mm00475834_m1), *Enpp1* (Mm01193761_m1), *Phospho1* (Mm07301398_m1), *Ank* (Mm00445040_m1), *Mepe* (Mm02525159_s1), *Phex* (Mm00448119_m1), *Fgf23* (Mm00445621_m1), *Fgfr1* (Mm00438930_m1), *Osteopontin* (Mm00436767_m1), *Dmp1* (Mm01208363_m1), and *Cathepsin B* (Mm01310506_m1). These gene expressions were detected using the StepOne Real-Time PCR System (Applied Biosystems), and they were normalized to *Gapdh* (Mm99999915_g1) expression using the 2ΔΔCt method.

### 2.9. Statistical Analysis

All values are presented as mean ± standard error. All statistical analyses were carried out using BellCurve for Excel version 2.00 (Social Survey Research Information Co., Ltd., Tokyo, Japan) and analyzed for statistical significance by Student’s t-test, which is one of the parametric tests. A *p*-value < 0.05 was considered significant. Additionally, power values were estimated by a power analysis.

## 3. Results

### 3.1. Metaphyseal Bone Mineralization and Immunolocalization of PHOSPHO1, TNALP, and ENPP1

The bone volume of the femoral metaphyses was higher in specimens administered with PTH two times and four times than in the control specimens (Figure 1A–C). Von Kossa staining showed a thick zone of an incompletely mineralized osteoid in the PTH-administered trabeculae but not in the control counterparts (Figure 1D–F). Additionally, the MAR index was significantly higher in specimens administered with PTH four times than in the control specimens (2.19 ± 0.05 vs. 1.47 ± 0.12, *p <* 0.01, 95% power) (Figure 1G). There was no significant difference in the serum Ca concentration, but the serum Pi concentration was significantly lower in PTH-administered specimens compared to the control specimens (8.56 ± 0.51 vs. 10.46 ± 0.66, *p <* 0.05, 63% power) (Figure 1H).

The areas positive for PHOSPHO1 appeared to expand after PTH administration (Figure 2A–C), and PHOSPHO1 immunoreactivity was mainly localized on the surface of mature osteoblasts in both the control and PTH-administered specimens (Figure 2D–F). The immunoreactivity of TNALP, which hydrolyzes PPi to Pi, was markedly intense in PTH-administered metaphyses when compared with the findings in the control counterparts (Figure 3A–C). At higher magnification, not only osteoblasts but also preosteoblastic cells appeared to show intense TNALP immunoreactivity (Figure 3D–F). ENPP1, which provides PPi, was broadly observed in osteoblasts in the PTH-administered metaphyseal trabeculae, whereas it was slightly observed in osteoblasts in the control counterparts (Figure 3G–L).

Double immunofluorescent analysis demonstrated that, in control specimens, ENPP1 was mainly localized throughout the cytoplasmic region of osteoblasts and that TNALP was roughly observed on the cell membrane (Figure 4A–D). On the other hand, in PTH-administered specimens, TNALP and ENPP1 seemingly overlapped in osteoblastic cells, although some osteoblastic cells showed intense ENPP1 reactivity (Figure 4E–H).

### 3.2. Immunolocalization of FGF23, MEPE, and Osteopontin in the Bone Matrix

We examined the immunolocalization of FGF23, MEPE, and osteopontin, which are related to the PHEX/SIBLING family axis. FGF23 immunoreactivity was detected in osteocytes in the control femora but was detected in both osteoblasts and osteocytes in the PTH-administered counterparts (Figure 5A–C). MEPE was hardly observed in the control femora but was evident in some osteocytes in the PTH-administered counterparts (Figure 5D–F). There appeared to be many osteopontin-immunopositive lines that were identical to cement lines after PTH administration (Figure 5G–I). There was no apparent difference in the immunolocalization of FGF23, osteopontin, or MEPE between the two times and four times per day regimens of PTH administration.

### 3.3. Gene Expressions of Tnalp, Enpp1, Phospho1, Ank, Mepe, Phex, Fgf23, Fgfr1, αKlotho, Osteopontin, Dmp1, and Cathepsin B

We examined the gene expressions of *Tnalp (Alpl), Enpp1, Phospho1, Ank, Mepe, Phex, Fgf23, Fgfr1, αKlotho, Osteopontin, Dmp1,* and *Cathepsin B* using RT-PCR and/or real-time PCR. All genes tended to be elevated on assessment with RT-PCR (Figure 6). Therefore, we quantified the expressions of these genes in control and PTH-administered specimens using real-time PCR (Table 1). The indices of *Tnalp (Alpl), Enpp1, Phospho1, Ank, Mepe, Phex, Fgf23, Fgfr1, Osteopontin,* and *Dmp1* were 2.7 (*p* < 0.01, 100% power), 1.8 (*p* < 0.01, 93% power), 2.8 (*p* < 0.01, 100% power), 1.3 (*p* < 0.01, 100% power), 2.0 (*p* < 0.01, 100% power), 4.4 (*p* < 0.01, 100% power), 1.4 (*p* < 0.01, 69% power), 2.2 (*p* < 0.01, 100% power), 2.5 (*p* < 0.01, 98% power), and 2.6 (*p* < 0.01, 100% power) times higher in specimens administered four times with PTH, respectively, compared to the control specimens (Table 1). However, the index of *Cathepsin B* was 0.9 times (*p* < 0.05, 63% power) lower in specimens administered four times with PTH compared with the findings in the control specimens (Table 1).

## 4. Discussion

The present study attempted to histochemically elucidate the immunolocalization and gene expressions of enzymes and membrane transporters involved in matrix vesicle-mediated mineralization, as well as the regulation of mineralization by the PHEX/SIBLING family axis after PTH administration. As a result, it seems likely that PTH would accelerate mineralization, as seen in an elevated MAR, and maintain a balance between promoted matrix synthesis and bone mineralization, presumably by mediating TNALP/ENPP1 cooperation and stimulating PHEX/SIBLING family expression. Here, we have employed normal young adult mice based on our previous studies [36,37] to evaluate the PTH effects on bone mineralization in a healthy state. However, it is imperative to examine the therapeutic effects of PTH on bone mineralization using osteoporotic models such as ovariectomized mice/rats in the future.

PTH systemically increases the serum Ca concentration by increasing bone turnover and Ca reabsorption in the kidneys and stimulating 1α-hydroxylase activity in the renal proximal tubules. However, the therapeutic administration of PTH did not markedly increase serum Ca due to the Ca secretion from the kidney. Meanwhile, our study showed that PTH reduced the serum Pi concentration. Despite the varied effects of PTH on serum Ca and Pi levels, the finely tuned activities of enzymes and membrane transporters associated with the micro-environment inside the matrix vesicle are more important than the Ca^2+^ level in serum.

TNALP expression and the areas of TNALP-immunoreactive osteoblasts/preosteoblasts markedly increased, and, consistently, PHOSPHO1 expression was more evident in the PTH-administered trabeculae. However, the distribution patterns of TNALP and PHOSPHO1 did not change after PTH administration. TNALP was maintained in both preosteoblastic cells and mature osteoblasts, whereas PHOSPHO1 was localized on the surface of mature osteoblasts even after PTH administration. On the other hand, some osteoblastic cells showed relatively intense ENPP1-positivity, presumably implying that these ENPP1-intense osteoblastic cells can regulate mineralization by synthesizing abundant PPi to inhibit excess mineralization. This appears reasonable when considering reports that extracellular PPi derived from ENPP1 activities inhibited mineralization by preventing the overgrowth of crystals [15,16,17,18]. Therefore, it appears likely that PTH might elevate the expressions of TNALP and PHOSPHO1 to facilitate mineralization and might simultaneously promote ENPP1 expression to regulate excessive mineralization. However, further examinations are necessary to identify which cell types show intense ENPP1 immunoreactivity.

Interestingly, not only osteocytes but also osteoblasts showed intense FGF23 immunoreactivity after PTH administration (Figure 5A–C). Our previous report localized FGF23 in osteoblasts and osteocytes in the embryonic and neonatal bones, indicating that FGF23 signaling might be involved in bone formation rather than in the systemic regulation of serum Pi at these developmental stages [28]. Recent reports have mentioned that osteoblast/osteocyte-derived FGF23 inhibited PHEX expression by mediating FGFR1c/αklotho in an autocrine manner [29] and, also, that FGF23 synthesized by osteocytes can reduce TNALP expression by mediating FGFR3, which is independent of the FGF23/αklotho axis [46]. These findings indicate an alternative pathway of FGF23 signaling in bone without the mediation of the αklotho/FGFR1c receptor in the kidneys.

Regarding the effects of the Phex/SIBLING family on mineralization, SIBLING family genes such as Mepe and osteopontin have been thought to be hydrolyzed by cathepsin B to generate ASARM peptides [47,48]. The serine residues of ASARM peptides are then phosphorylated to form pASARM, which can bind hydroxyapatite crystals, consequently inhibiting mineral crystal growth. On the contrary, PHEX has been documented to bind to the SIBLING family and block the hydrolysis of cathepsin B, consequently avoiding the synthesis of ASARM peptides [48,49]. In this study, the expression of the *Phex* gene markedly increased by 4.4 times, and the expression of the *Cathepsin B* gene reduced by 0.9 times in PTH-administered specimens compared with the findings in the control specimens. Considering that PHEX binds to the SIBLING family to block hydrolysis by cathepsin B, the increased expression of *Phex* and reduced expression of *Cathepsin B* could allow further mineralization in bone by reducing the production of phosphorylated ASARM, which inhibits mineralization. As MEPE and osteopontin were localized in the regions of osteocytes and cement lines, respectively, further mineralization by the PHEX/SIBLING family axis might occur not in the osteoid but in the deep portion of the bone matrix.

The administration of recombinant human PTH (1–34), teriparatide, has reportedly elevated BMD [50,51,52]. The BMD of osteoporotic patients has been measured by dual-energy X-ray absorptiometry at a macroscale, and a few studies that conducted measurements at a microscale have shown the gene expressions and immunolocalization of bone samples of PTH-treated osteoporotic patients. The enzymes/membrane transporters examined in this study play an important role in bone mineralization in humans. Indeed, mutations in TNALP result in hypophosphatasia, which can be categorized into severe, moderate, and mild forms by their genetic characteristics. Severe forms of hypophosphatasia with perinatal and infantile severity are recessively inherited, whereas moderate hypophosphatasia may be inherited dominantly or recessively [53]. The lack of ENPP1 was found to be associated with the spontaneous mineralization of infantile arteries and periarticular regions [54,55]. Further, mutated ANK that acts as a plasma membrane PPi channel induces the elevated deposition of calcium pyrophosphate crystals in the synovial fluid in humans [56]. It is well known that PHEX mutations induced the over-expression of FGF23, leading to severe hypophosphatemia, also known, specifically, as X-linked hypophosphatemia [57]. Thus, bone mineralization appears finely tuned by the TNALP/ENPP1 and PHEX/SIBLING family.

## 5. Conclusions

PTH seems to facilitate the gene expressions and protein levels of TNALP, PHOSPHO1, and ENPP1 to induce mineralization in the osteoid while maintaining a balance with the accelerated synthesis of matrix proteins and mineralization by means of stimulating the expressions of PHEX and the SIBLING family but reducing the expression of cathepsin B, consequently inducing further mineralization in the deep portion of the bone matrix.

## Figures and Tables

**Figure 1 medicina-59-01179-f001:**
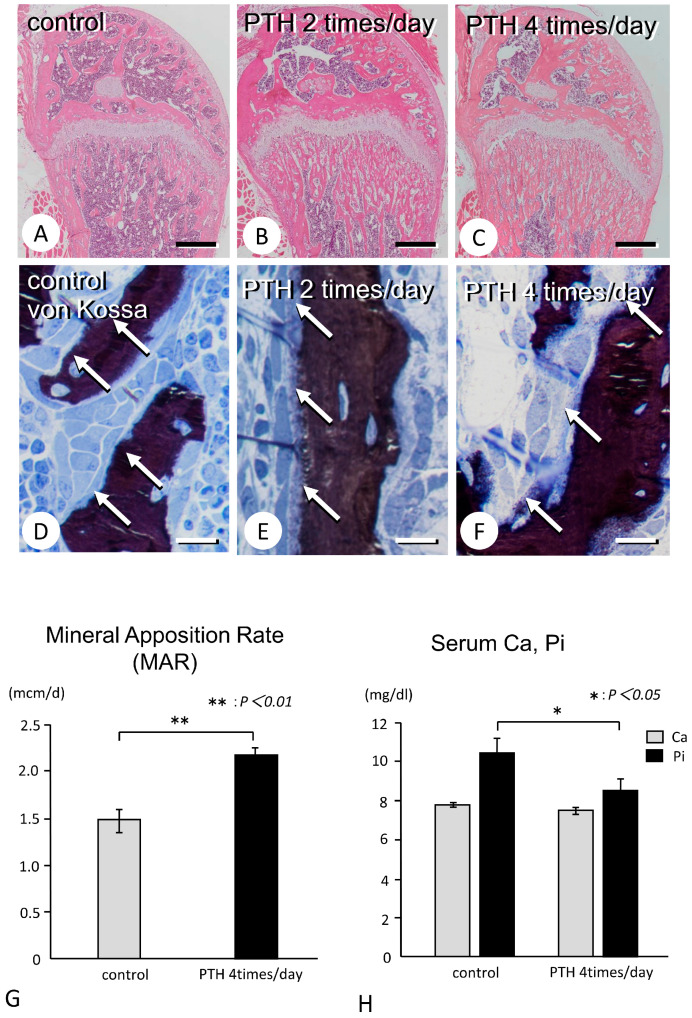
Increased trabecular bone and MAR in PTH-administered femora. Panels (**A**–**F**) are obtained from the control femora and femora administered with PTH in the regimen of 2 times/day and 4 times/day. According to the frequency of PTH administration, the metaphyseal trabeculae bone volume seems to be increased (**A**–**C**), and thick osteoid (white arrows) can be seen (**D**–**F**). Panel (**G**) shows the statistical analysis of MAR. (**H**) shows the statistical analysis of serum Ca and Pi. Panel H is a graph representing the statistical analysis on the serum Ca and Pi between the control and PTH-administered specimens. Bars: (**A**–**C**): 500 µm; (**D**–**F**): 10 µm.

**Figure 2 medicina-59-01179-f002:**
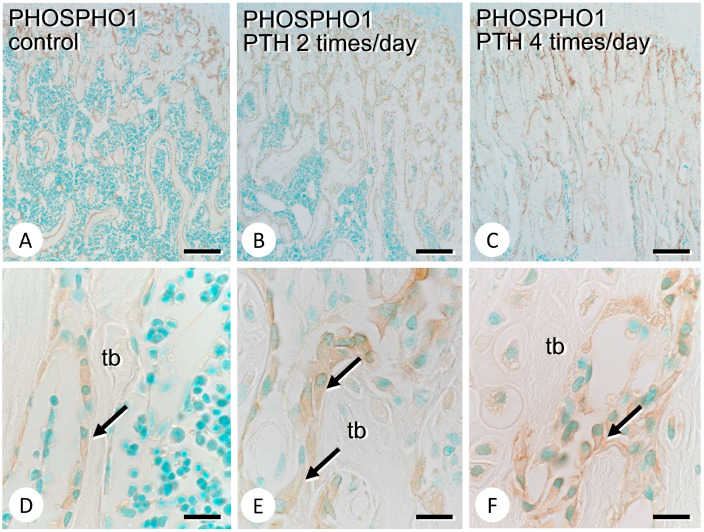
Immunolocalization of PHOSPHO1 in the control and PTH-administered femora. Panels (**A**–**C**) and (**D**–**F**) are low- and high-magnification images of immunostaining of PHOSPHO1 in the metaphyses of the control (**A**,**D**) and PTH-administered (**B**,**C**,**E**,**F**) femora, respectively. Note that PHOSPHO1-immunoreactivity is mainly seen on the cell surface of mature osteoblasts rather than preosteoblastic cells (arrows in **D**–**F**). tb: trabecula. Bars: (**A**–**C**): 100 µm; (**D**–**F**): 10 µm.

**Figure 3 medicina-59-01179-f003:**
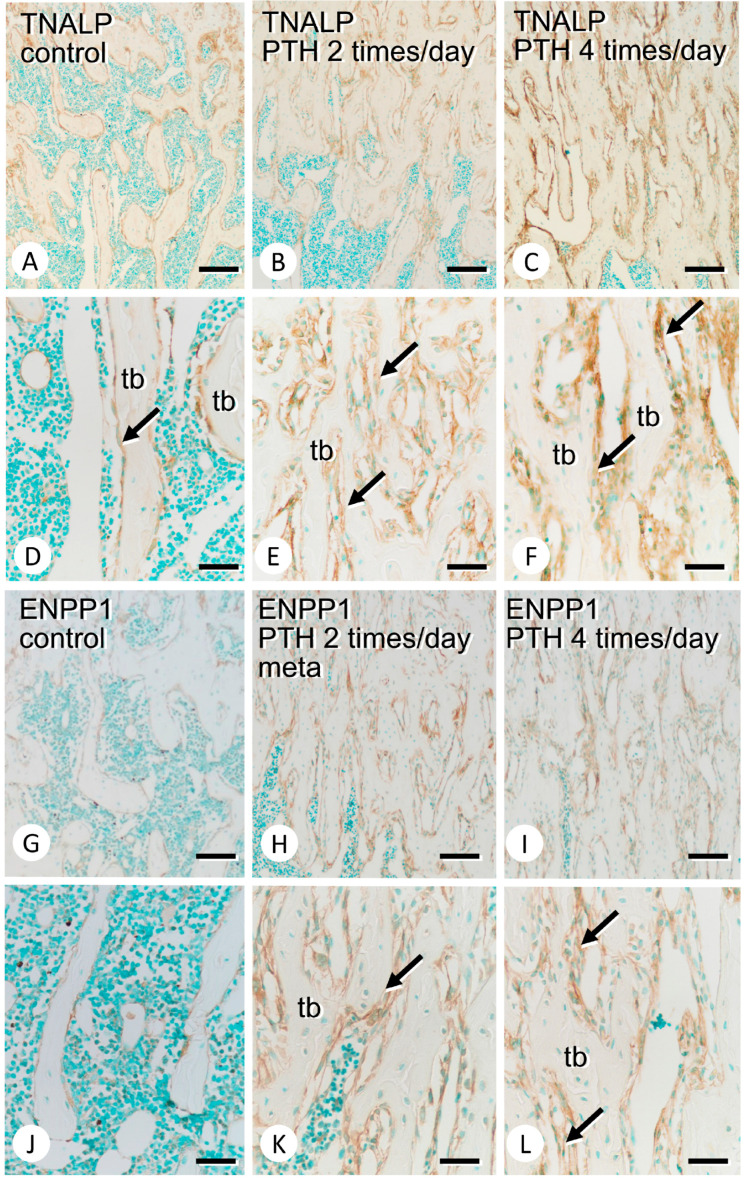
Immunolocalization of TNALP and ENPP1 in the control and PTH-administered femora. Panels (**A**–**C**) and (**D**–**F**) are low- and high-magnification images of immunostaining of TNALP in the metaphyses of the control (**A**,**D**) and PTH-administered (**B**,**C**,**E**,**F**) femora, respectively. On the other hand, panels (**G**–**I**) and (**J**–**L**) are low- and high-magnification images of immunostaining of ENPP1 in the metaphyses of the control (**G**,**J**) and PTH-administered (**H**,**I**,**K**,**L**) femora, respectively. After PTH administration, intense immunoreactivities of TNALP (arrows in **E**,**F**) and ENPP1 (arrows in **K**,**L**) can be seen. tb: trabecula. Bars: (**A**–**C**,**G**–**I**): 100 µm; (**D**–**F**,**J**–**L**): 30 µm.

**Figure 4 medicina-59-01179-f004:**
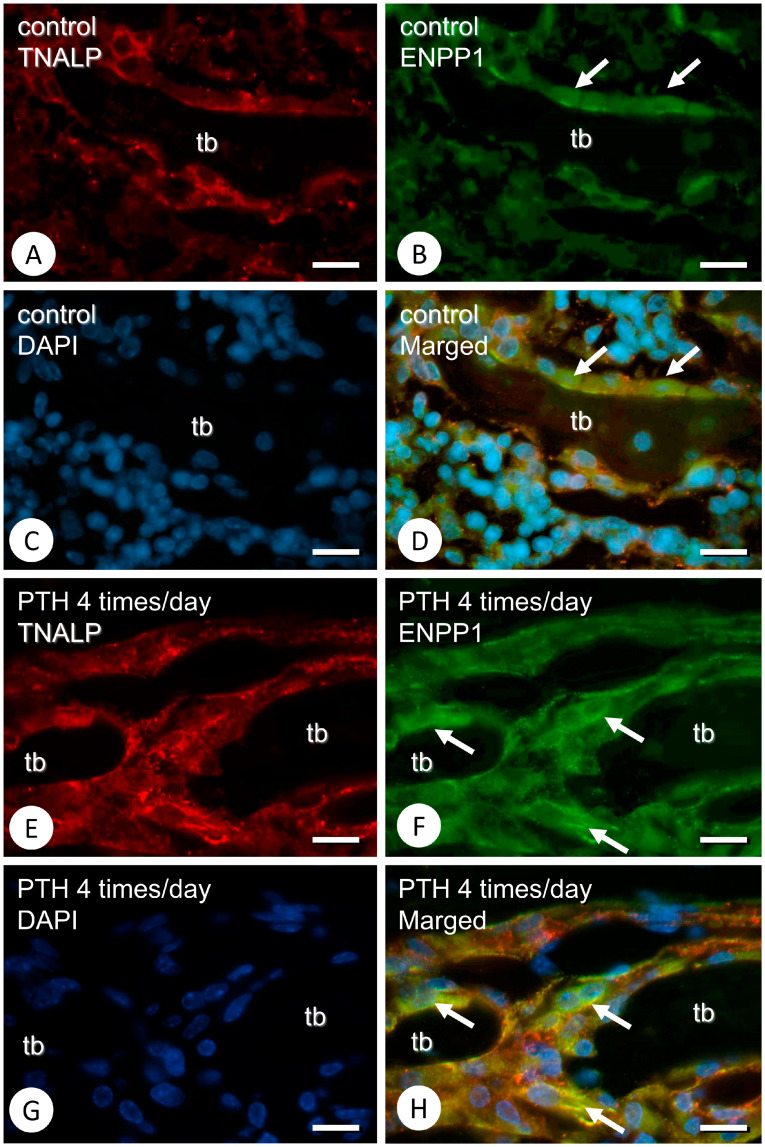
Immunofluorescent analysis of TNALP and ENPP1 in the control and PTH-administered femora. Panels (**A**–**D**) show immunolocalization of TNALP and ENPP1 in the control specimens, while panels (**E**–**H**) are those obtained from the PTH-administered samples. Panels (**A**,**B**,**D**–**F**,**H**) demonstrate TNALP- and ENPP1-immunolocalization and the merged images of TNALP and ENPP1. Although the uniform distribution of ENPP1 can be seen throughout the control osteoblasts’ cytoplasm (arrows in **B**,**D**), some osteoblastic cells with intense immunoreactivity of ENPP1 are observed after PTH administration (arrows in **F**,**H**). tb: trabecula. Bars (**A**–**H**): 10 µm.

**Figure 5 medicina-59-01179-f005:**
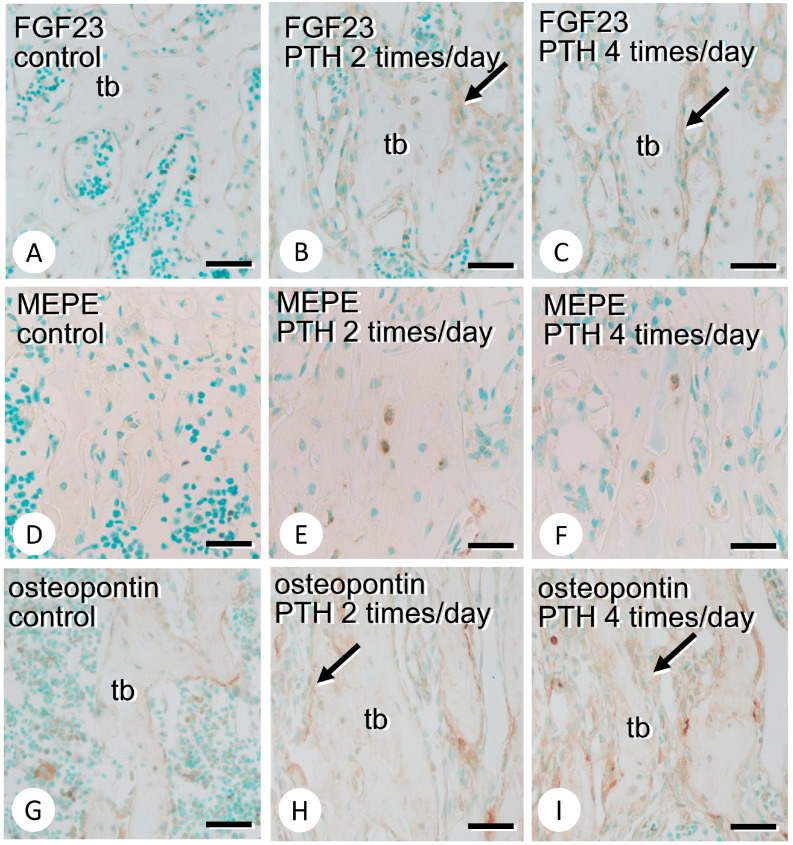
Immunolocalization of FGF23, MEPE, and osteopontin in the control and PTH-administered metaphyseal trabeculae. Panels (**A**–**C**,**D**–**F**,**G**–**I**) are immunodetection of FGF23, MEPE, and osteopontin, respectively. Panels (**A**,**D**,**G**) show the control metaphyses, while panels (**B**,**E**,**H**) and (**C**,**F**,**I**) are from PTH-administered trabeculae with regimens of 2 times or 4 times/day, respectively. FGF23 is observed mainly in osteocytes of control specimens (**A**), while FGF23 can be seen not only in osteocytes but also in osteoblasts after the PTH administration (arrows in **B**,**C**). MEPE is observable in osteocytes after the PTH treatment (**E**,**F**), while it is hardly detected in the control specimens (**D**). Note intense osteopontin-immunoreactive cement lines (arrows in **H**,**I**) in the PTH-administered specimens compared to the control counterparts (**G**). tb: trabecula. Bars (**A**–**I**): 30 µm.

**Figure 6 medicina-59-01179-f006:**
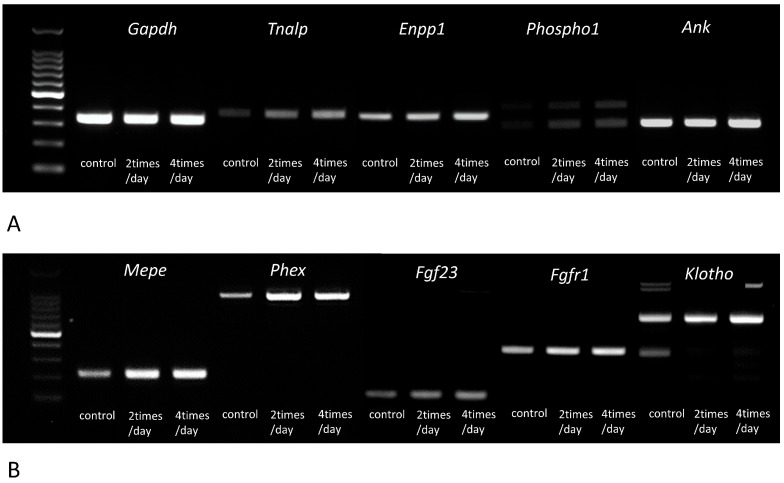
The gene expression profiles of *Tnalp*, *Enpp1*, *Phospho1*, *Ank*, *Mepe*, *Phex*, *Fgf23*, *Fgfr1*, and *klotho* assessed by RT-PCR in the control and PTH-administered bones. Panel (**A**) reveals RT-PCR results indicating the gene expressions of *Tnalp*, *Enpp1*, *Phospho1*, and *Ank*, while panel (**B**) displays the expressions of *Mepe*, *Phex*, *Fgf23*, *Fgfr1*, and *klotho* examined by RT-PCR.

**Table 1 medicina-59-01179-t001:** The gene expression profiles of *Tnalp, Enpp1, Phospho1, Ank, Mepe, Phex, Fgf23, Fgfr1, Osteopontin, Dmp-1*, and *Cathepsin B* by real-time PCR in the control and PTH-administered bones. Real-time PCR analysis reveals that only the expression of *Cathepsin B* is significantly reduced after the PTH administration, while the other genes showed elevated expression.

	Gene Expression (Mean ± SE)		*p*-Value	Power (%)
	Control	PTH 4 Times/Day		
*Tnalp (Alpl)*	1.00 ± 0.13	2.74 ± 0.19	*p* < 0.01	100
*Enpp1*	1.00 ± 0.14	1.76 ± 0.11	*p* < 0.01	93
*Phosoho1*	1.00 ± 0.13	2.79 ± 0.13	*p* < 0.01	100
*Ank*	1.00 ± 0.02	1.28 ± 0.03	*p* < 0.01	100
*Mepe*	1.00 ± 0.10	1.98 ± 0.17	*p* < 0.01	100
*Phex*	1.00 ± 0.14	4.37 ± 0.45	*p* < 0.01	100
*Fgf23*	1.00 ± 0.03	1.38 ± 0.11	*p* < 0.01	69
*Fgfr1*	1.00 ± 0.02	2.18 ± 0.06	*p* < 0.01	100
*Osteopontin*	1.00 ± 0.12	2.45 ± 0.28	*p* < 0.01	98
*Dmp1*	1.00 ± 0.10	2.65 ± 0.18	*p* < 0.01	100
*Cathepsinb*	1.00 ± 0.01	0.92 ± 0.03	*p* < 0.05	63

## Data Availability

Not applicable.

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
