# Peer review of "Immunolocalization of Enzymes/Membrane Transporters Related to Bone Mineralization in the Metaphyses of the Long Bones of Parathyroid-Hormone-Administered Mice"

_medicina, 2023, doi:10.3390/medicina59061179_

Round 1
Reviewer 1 Report
1. Please providing rationales why authors focused on bone mineralization and PTH in introduction section? For novel findings or potential therapeutic strategy? And what's the relationshop betweenbone mineralization and PTH?
2. Since Ca2+ was not changed after PTH treatment, was there any novelty presented in this study? If so, please discissed or emphasized on this issue in the main text.
3. Authors should describe more detail in "2.1. Animals" of Materials and Methods, following are some questions for this section, (1). why normal animals were used to study this topic? Please provide sifficient rationales for doing this. (2). Could the hypothesis be applied to desease model ? If so, it was more valuable, wasn't it? (3). There were lots of enzymes /membrane transporters that associated with mineralization, why authors focused on those, especially, the healthy animals were used in this experimental layout?
4. Would authors try to focus on another minerals while Ca2+ did not show expected change? Therefore, the scientific significance of this study was weak.
Author Response
Our Responses to the Reviewer #1
Thank you for your invaluable suggestions that helped us improve the quality of our manuscript. Please find below our point-by-point responses to your comments. The corresponding changes have been incorporated into the main manuscript.
Reviewer’s comments
- Please providing rationales why authors focused on bone mineralization and PTH in introduction section? For novel findings or potential therapeutic strategy? And what's the relationship between bone mineralization and PTH?
Our response
Thank you for your invaluable comments. As suggested by the reviewer, we have clearly described the rationale of the current study in the introduction section of the revised manuscript. Although the intermittent administration of PTH is used for osteoporotic treatment, whether it elevates the expression of genes encoding the enzymes and membrane transporters associated with bone mineralization is still veiled. Therefore, studying the influence of PTH on the expression of these genes would provide novel insight into bone mineralization mechanisms. In line with this, the following sentences have been added in the first paragraph of the introduction section.
Introduction, page 2, line 88-93
Parathyroid hormone (PTH) (1-34) is a calcitropic hormone that stimulates osteoclastic bone resorption and osteoblastic bone formation. Although the intermittent administration of hPTH (1-34) increases bone volume [36, 37], whether it elevates the expression of genes encoding enzymes and membrane transporters associated with bone mineralization is still veiled. Therefore, studying the influence of PTH in vivo on the expression of these genes would provide novel insight into bone mineralization mechanisms.
Reviewer’s comments
- Since Ca2+ was not changed after PTH treatment, was there any novelty presented in this study? If so, please discussed or emphasized on this issue in the main text.
Our response
The authors are thankful for the reviewer’s suggestion. The reviewer has pointed out that the serum Ca2+ was not changed after PTH treatment. The serum Ca2+ concentration is maintained due to Ca2+ secretion in the kidney, despite the frequent administration of hPTH (1-34) to the mice, which implies that the finally-tuned activities of the enzymes and membrane transporters associated with the micro-environment inside the matrix vesicle are more important than the Ca2+ level in serum. Therefore, we have added the following sentences in the discussion section of the revised manuscript.
Discussion, page 12, line 369-376
PTH systemically increases the serum Ca concentration by increasing bone turnover and Ca reabsorption in the kidneys and stimulating 1a-hydroxylase activity in the renal proximal tubules. However, the therapeutic administration of PTH did not markedly increase serum Ca due to the Ca secretion from the kidney. Meanwhile, our study showed that PTH reduced the serum Pi concentration. Despite the varied effects of PTH on serum Ca and Pi levels, the finally-tuned activities of enzymes and membrane transporters associated with the micro-environment inside the matrix vesicle are more important than the Ca2+ level in serum.
Reviewer’s comments
- Authors should describe more detail in "2.1. Animals" of Materials and Methods, following are some questions for this section, (1). why normal animals were used to study this topic? Please provide sifficient rationales for doing this. (2). Could the hypothesis be applied to desease model ? If so, it was more valuable, wasn't it? (3). There were lots of enzymes /membrane transporters that associated with mineralization, why authors focused on those, especially, the healthy animals were used in this experimental layout?
Our response
Our response to the comments (1) and (2)
To address the first question by the reviewer, we have employed normal young adult mice to evaluate the effects of PTH on bone mineralization. The current approach was designed following our previous study [36, 37]. We agree that the reviewer’s concern regarding the PTH action on bone mineralization as therapeutic effects in osteoporotic patients is valid. The most common rodent models mimicking post-menopausal osteoporosis, which is a primary and major osteoporosis, are ovariectomized mice and rats. However, the examination of the PTH action during estrogen deficiency seems to be complicated; therefore, to begin with, we sought to examine the PTH effects on bone mineralization in a normal healthy state. We believe that the reviewer’s comment is important as a perspective of the current study and can be considered while designing future studies.; therefore, we have added the following sentences in the discussion section.
Discussion, page 12, line 364-368
Here, we have employed normal young adult mice based on our previous studies [36, 37] to evaluate the PTH effects on bone mineralization in a healthy state. However, it is imperative to examine the therapeutic effects of PTH on bone mineralization using osteoporotic models such as ovariectomized mice/rats in the future.
Our response to the comments (3)
As pointed out by the reviewer, several candidate enzymes and membrane transporters might be involved in bone mineralization. However, according to the available literature, many are still not functionally characterized. We would like to highlight that the enzymes and membrane transporters mentioned in the current study are precisely involved in matrix vesicle-mediated bone mineralization and its regulation.
Reviewer’s comments
- Would authors try to focus on another minerals while Ca2+ did not show expected change? Therefore, the scientific significance of this study was weak.
Our response
Thank you for your valuable comment. Bone minerals are composed of crystalline calcium phosphates; therefore, the action of enzymes and membrane transporters enabling the Ca2+ and Pi supplementation into the matrix vesicles is important. Other minerals, such as magnesium, zinc, etc., are categorized as minor minerals, which are not included in hydroxyapatite. However, your comment made us realize that it would be interesting for the readers to have a brief idea about matrix vesicle-mediated mineralization and the importance of Pi supplementation in matrix vesicles in the current article. Therefore, we have added the following sentences.
Introduction, page 1-2, line 39-57
Bone matrix primarily comprises calcium phosphate crystals (hydroxyapatite) and type I collagen; therefore, bone formation can be divided into two aspects: the synthesis of organic bone matrices, including collagens and non-collagenous proteins, and the mineralization of bone matrix. Bone mineralization, i.e., calcium phosphate crystals deposited onto the collagen fibrils, provides hardness, while type I collagen gives flexibility to the bone. Bone mineralization is biologically regulated by osteoblasts that secrete matrix vesicles, small extracellular vesicles, which initiate the mineralization process in bones, i.e., matrix vesicle-mediated mineralization [1–5]. To allow matrix vesicle-mediated mineralization, the inflow of abundant calcium ions (Ca2+) and inorganic monophosphate ions or Pi ions (PO43-) into the matrix vesicles is imperative to nucleate calcium phosphate crystals. The crystalline calcium phosphates grow inside the matrix vesicles and penetrate the plasma membranes of the vesicles to form mineralized nodules, which are globular assemblies of fine needle-like calcium phosphate crystals [1, 2, 5].
An important aspect of bone mineralization is the biological action on nucleation and growth of crystalline calcium phosphates and the Ca2+ and PO43- supplementation. There are numerous free Ca2+ ions in tissue fluid surrounding the matrix vesicles to support the direct inflow of Ca2+ into the vesicles by Ca2+-ATPase [2]. In addition, mature osteoblasts, and matrix vesicles have several membrane transporters and enzymes involved in the local Pi supplementation.
Finally, we would like to express our sincere appreciation to the Reviewer. We believe that your invaluable suggestions helped us greatly improve our manuscript.

Reviewer 2 Report
The authors present an interesting study on the immunolocalization of enzymes/membrane transporters related to bone mineralization.
Minor review is required, following these suggestions:
1) Introduction
The introduction lacks in presenting a general overview of bone tissue complexity, that ranges from the macro- up to the micro-scale (see https://doi.org/10.1016/j.engfracmech.2022.108582, https://doi.org/10.1038/bonekey.2015.112, https://doi.org/10.3390/ma14051240). This is a core point to introduce the need for a deeper focus on bone micro-scale.
2) Materials and Methods.
2.1 Do the authors perform a power analysis to check the significance level of their assumptions? Please provide it.
2.9 Provide more details on the statistical analysis. Did you perfor parametric or non parametric tests? Which kind of tests? Please provide further motivations and details.
4) Discussion
The discussion sectionshould perform deeper comparisons between authors' results and state-of-the-art analyses. As an example, what about translating the results from rat model to human one? Are there literature data providing information about this point?
Author Response
Our responses to the reviewer #2
We appreciate the invaluable suggestions provided by the reviewer. We agree with your suggestions. Kindly see the improvements that were made following your comments. We believe that our paper has been significantly enhanced because of your helpful suggestions. Thank you very much for your important comments.
Reviewer’s general comments
The authors present an interesting study on the immunolocalization of enzymes/membrane transporters related to bone mineralization.
Minor review is required, following these suggestions:
Reviewer’s specific comment #1
1) Introduction
The introduction lacks in presenting a general overview of bone tissue complexity, that ranges from the macro- up to the micro-scale (see https://doi.org/10.1016/j.engfracmech.2022.108582, https://doi.org/10.1038/bonekey.2015.112, https://doi.org/10.3390/ma14051240). This is a core point to introduce the need for a deeper focus on bone micro-scale.
Our response
We agree with the reviewer’s suggestion. Since this study focuses on bone mineralization, we added the sentences below on page 1-2, line 39-57 following the reviewer’s comment. In addition, the references mentioned by the reviewer are related to the complexity of multiscale damage processes and disease-related deterioration in bone’s mechanical properties and are important since mineralization is required for bone’s mechanical properties. Therefore, we also added the below-given sentences on page 2-3, line 94-109.
Introduction, page 1-2, line 39-57
Bone matrix primarily comprises calcium phosphate crystals (hydroxyapatite) and type I collagen; therefore, bone formation can be divided into two aspects: the synthesis of organic bone matrices, including collagens and non-collagenous proteins, and the mineralization of bone matrix. Bone mineralization, i.e., calcium phosphate crystals deposited onto the collagen fibrils, provides hardness, while type I collagen gives flexibility to the bone. Bone mineralization is biologically regulated by osteoblasts that secrete matrix vesicles, small extracellular vesicles, which initiate the mineralization process in bones, i.e., matrix vesicle-mediated mineralization [1-5]. To allow matrix vesicle-mediated mineralization, the inflow of abundant calcium ions (Ca2+) and inorganic monophosphate or Pi ions (PO43-) into the matrix vesicles is imperative to nucleate calcium phosphate crystals. The crystalline calcium phosphates grow inside the matrix vesicles and penetrate the plasma membranes of the vesicles to form mineralized nodules, which are globular assemblies of fine needle-like calcium phosphate crystals [1, 2, 5].
An important aspect of bone mineralization is the biological action on nucleation and growth of crystalline calcium phosphates and the Ca2+ and PO43- supplementation. There are numerous free Ca2+ in tissue fluid surrounding the matrix vesicles to support the direct inflow of Ca2+ into the vesicles by Ca2+-ATPase [2]. In addition, mature osteoblasts and matrix vesicles have several membrane transporters and enzymes involved in the local Pi supplementation.
Introduction, page 2-3, line 94-109
To estimate bone mineralization, microCT analyses were performed on mice and rats, which provided information regarding mineral contents and bone mineral density (BMD), as well as bone volume and geometrical structure not only at a macroscale of the whole bone but also each part of the bone. A long bone is roughly composed of an outer cortical bone and a spongy inner bone. The extremity of an adult long bone is the epiphyses, while young adult mice/rats contain growth plate cartilage between the proximal epiphyses and distal metaphyses. The region of midshaft is diaphysis featuring cortical bone. MicroCT analysis estimates mineral contents and BMD in these parts of long bones at a macroscale. However, to elucidate the PTH effects on the mineralization process and the enzymes/membrane transporters related to bone mineralization, it is necessary to employ microscale observation, localizing these enzymes/transporters on osteoblasts and osteocytes. As shown by Zimmermann et al. [38] and Buccino et al. [39, 40], it is reasonable to explore the complexity of multiscale damage processes that make fracture prediction and also significant to seek an understanding of the origins of disease-related deterioration in bone’s mechanical properties. Consistent with these reports, bone mineralization appears to be involved in the bone’s mechanical properties.
Reviewer’s specific comment #2
2) Materials and Methods.
2.1 Do the authors perform a power analysis to check the significance level of their assumptions? Please provide it.
Our responses
Thank you for your valuable comment. As suggested by the reviewer, we performed a power analysis for all the statistical analyses and added these data in the Results section of the revised manuscript. In addition, we changed Fig. 6 and Fig. 7 to Table 1 and figure legends.
According to the power analysis in our study, MAR and gene expressions of Tnalp, Enpp1, Phospho1, Ank. Mepe, Phex, Fgfr1, Osteopontin, and Dmp1 showed high values of power, while the concentrations of serum calcium, serum Pi, and the gene expressions of Fgf23 and Cathepsin b displayed low values of power (Fig.1). The sample size should be set to 80-90% power; however, the values of power widely varied among the analysis items in our study. Therefore, we presumed it might not be easy to determine the common sample size with the same power value for all items. In the current study, we employed six mice in each group since the previous animal experiments involved approximately 5-10 animals in each group to accommodate individual variation. When the same analysis was performed on three mice in each group, the power values were reduced in all the items (Fig. 2). In addition, all analyses of gene expression displayed approximately 80% power and statistical significance, although the power values of serum Ca and serum Pi didn’t show the suitable indices. For the reviewer’s reference, we would like to present these data as follows.
|
Fig. 1 The results of Student’s t-test and power test in this study (n= 6 in each group) |
|
Fig. 2 The results of Student’s t-test and power test in this study (n= 3 in each group) |
Results, page 5, line 252-257
Additionally, the MAR index was significantly higher in four times PTH-administered specimens than in the control (2.19 ± 0.05 vs. 1.47 ± 0.12, P < 0.01, 95% power) (Fig. 1G). There was no marked difference in the serum Ca concentration, but the serum Pi concentration was significantly lower in PTH-administered specimens than in control (8.56 ± 0.51 vs. 10.46 ± 0.66, P < 0.05, 63% power) (Fig. 1H).
Results, page 11, line 332-343
We examined the gene expressions of Tnalp (Alpl), Enpp1, Phospho1, Ank, Mepe, Phex, Fgf23, Fgfr1, aKlotho, Osteopontin, Dmp1, and Cathepsin B using RT-PCR and/or real-time PCR. All genes tended to be elevated on assessment with RT-PCR (Figs. 6). Therefore, we quantified the expressions of these genes in control and PTH-administered specimens using real-time PCR (Table 1). The indices of Tnalp (Alpl), Enpp1, Phospho1, Ank, Mepe, Phex, Fgf23, Fgfr1, Osteopontin, and Dmp1 were 2.7 (P < 0.01, 100% power), 1.8 (P < 0.01, 93% power), 2.8 (P < 0.01, 100% power), 1.3 (P < 0.01, 100% power), 2.0 (P < 0.01, 100% power), 4.4 (P < 0.01, 100% power), 1.4 (P < 0.01, 69% power), 2.2 (P < 0.01, 100% power), 2.5 (P < 0.01, 98% power), and 2.6 (P < 0.01, 100% power) times higher in four times PTH-administered specimens, respectively, as compared to the control specimens (Table 1). However, the index of Cathepsin B was 0.9 times (P < 0.05, 63% power) lower in four times PTH-administered specimens when compared with the findings in control specimens (Table 1).
Figure legends, page 11, line 345-348
Figure 6. The gene expression profiles of Tnalp, Enpp1, Phospho1, Ank, Mepe, Phex, Fgf23, Fgfr1, and klotho assessed by RT-PCR in the control and PTH-administered bones. Panel A reveals RT-PCR results indicating the gene expressions of Tnalp, Enpp1, Phospho1, and Ank, while panel B displays the expressions of Mepe, Phex, Fgf23, Fgfr1, and klotho examined by RT-PCR.
Figure legends, page 12, line 352-355
Table 1. The gene expression profiles of Tnalp, Enpp1, Phospho1, Ank, Mepe, Phex, Fgf23, Fgfr1, Osteopontin, Dmp-1, and Cathepsin B by real-time PCR in the control and PTH-administered bones. Real-time PCR analysis reveals that only the expression of Cathepsin B is significantly reduced after the PTH administration, while the other genes showed elevated expression.
Reviewer’s specific comment #3
2.9 Provide more details on the statistical analysis. Did you perfor parametric or non parametric tests? Which kind of tests? Please provide further motivations and details.
Our responses
Thank you for your valuable suggestions. We have performed the parametric test in our study and accordingly rephrased the M&M section in the revised manuscript as follows.
Materials and Methods, page 5, line 240-245
Statistical analysis
All values are presented as mean ± standard error. All statistical analyses were carried out using BellCurve for Excel version 2.00 (Social Survey Research Information Co., Ltd., Tokyo, Japan) and analyzed for statistical significance by Student’s t-test, which is one of the parametric tests. A P-value <0.05 was considered significant. Additionally, power values were estimated by a power analysis.
Reviewer’s specific comment #4
4) Discussion
The discussion section should perform deeper comparisons between authors' results and state-of-the-art analyses. As an example, what about translating the results from rat model to human one? Are there literature data providing information about this point?
Our responses
Thank you for your invaluable comments. Few studies have demonstrated immunolocalization of TNALP/ENPP1 and PHEX/SIBLING family in PTH-administered patients. However, we believe that the reviewer’s point is valid that these enzymes/transporters play pivotal roles in mineralization in humans. Therefore, we have described the following sentence in the revised manuscript.
Discussion, page 13, line 417-432
Administration of recombinant human PTH [1-34], teriparatide, has reportedly elevated BMD [50-52]. However, the BMD of osteoporotic patients has been measured by dual-energy X-ray absorptiometry at a macroscale. A few studies at a microscale have shown the gene expression and immunolocalization in bone samples of PTH-treated osteoporotic patients. The enzymes/membrane transporters examined in this study play an important role in bone mineralization in humans. Indeed, mutations in TNALP result in hypophosphatasia, which can be categorized into severe, moderate, and mild forms by their genetic characteristics. Severe forms of hypophosphatasia with perinatal and infantile severity are recessively inherited, whereas moderate hypophosphatasia may be inherited dominantly or recessively [53]. The lack of ENPP1 was found to be associated with the spontaneous mineralization of infantile arteries and periarticular regions [54, 55]. Further, mutated ANK that acts as a plasma-membrane PPi channel induces elevated deposition of calcium pyrophosphate crystals in the synovial fluid in humans [56]. It is well known that PHEX mutation induced the over-expression of FGF23, leading to severe hypophosphatemia, which is X-linked hypophosphatemia [57]. Thus, bone mineralization appears finely tuned by TNALP/ENPP1 and PHEX/SIBLIMNG family.
Finally, we would like to express our sincere appreciation to the reviewer #2. We believe that your invaluable suggestions have helped us significantly improve our manuscript.
